# Conformational heterogeneity of molecules physisorbed on a gold surface at room temperature

Mingu Kang [1,2], Hyunwoo Kim[3], Elham Oleiki [4], Yeonjeong Koo[1,2], Hyeongwoo Lee[1,2], Huitae Joo[2], Jinseong Choi[2], Taeyong Eom[5], Geunsik Lee [4], Yung Doug Suh[3,4,6 ✉] & Kyoung-Duck Park[1,2 ✉]

A quantitative single-molecule tip-enhanced Raman spectroscopy (TERS) study at room temperature remained a challenge due to the rapid structural dynamics of molecules exposed to air. Here, we demonstrate the hyperspectral TERS imaging of single or a few brilliant cresyl blue (BCB) molecules at room temperature, along with quantitative spectral analyses. Robust chemical imaging is enabled by the freeze-frame approach using a thin $Al_2O_3$ capping layer, which suppresses spectral diffusions and inhibits chemical reactions and contamination in air. For the molecules resolved spatially in the TERS image, a clear Raman peak variation up to 7.5 cm$^{-1}$ is observed, which cannot be found in molecular ensembles. From density functional theory-based quantitative analyses of the varied TERS peaks, we reveal the conformational heterogeneity at the single-molecule level. This work provides a facile way to investigate the single-molecule properties in interacting media, expanding the scope of single-molecule vibrational spectroscopy studies.

---

[1] Department of Physics, Pohang University of Science and Technology (POSTECH), 77, Cheongam-ro, Pohang 37673, Republic of Korea. [2] Department of Physics, Ulsan National Institute of Science and Technology (UNIST), 50, UNIST-gil, Ulsan 44919, Republic of Korea. [3] Laboratory for Advanced Molecular Probing (LAMP), Korea Research Institute of Chemical Technology (KRICT), 141, Gajeong-ro, Daejeon 34114, Republic of Korea. [4] Department of Chemistry, Ulsan National Institute of Science and Technology (UNIST), 50, UNIST-gil, Ulsan 44919, Republic of Korea. [5] Thin Film Materials Research Center, Korea Research Institute of Chemical Technology (KRICT), 141, Gajeong-ro, Daejeon 34114, Republic of Korea. [6] School of Energy and Chemical Engineering, Ulsan National Institute of Science and Technology (UNIST), 50, UNIST-gil, Ulsan 44919, Republic of Korea. ✉email: ydsuh@unist.ac.kr; parklab@postech.ac.kr

Observations of single molecules in different chemical environments[1–11] are enabled via surface-enhanced Raman scattering (SERS) or tip-enhanced Raman spectroscopy (TERS), based on their characteristic spectral "fingerprint"[12–15]. In general, SERS provides a larger enhancement factor in single-molecule detection compared to TERS[16] which gives access to extremely weak vibrational responses of single molecules[17]. On the other hand, TERS allows us to probe even individual chemical bonds in a single molecule[18–21] using a strongly localized optical field at the plasmonic nano-tip[22–24], controlled by scanning probe microscopy approaches[16,25].

Specifically, these experiments revealed the conformational heterogeneity, intramolecular coupling, vibrational dephasing, and molecular motion of single molecules at cryogenic temperatures under ultrahigh vacuum environments[5,18,26–28]. The extreme experimental conditions are advantageous to reduce rotational and spectral diffusions of single molecules and prevent contamination of tips from a surrounding medium. On the other hand, the cryogenic TERS setup cannot be widely deployed because its configuration is highly complicated and the level of difficulty for experiments is also very high. Moreover, performing single-molecule TERS experiments at room temperature is necessarily required to investigate the molecular functions and interactions with respect to chemical environments, such as temperature and atmospheric condition[5,29,30].

In particular, understanding the conformational heterogeneity of single molecules in the non-equilibrium state is highly desirable because it can address many fundamental questions regarding the structure and function of many biological systems[31–35], such as protein folding[36,37] and RNA dynamics[38–40]. Previously, a few TERS groups detected single molecules at room temperature[30,41,42], yet only limited molecular properties were characterized due to the rapid structural dynamics of molecules exposed to air. Therefore, a systematic approach for robust single-molecule TERS experiments at room temperature is highly desirable.

Here, we present a room-temperature freeze-frame approach for single-molecule TERS. To capture the single molecules, we deposit an atomically thin dielectric capping layer (0.5 nm thick $Al_2O_3$) onto the molecules on the metal substrate. The freeze-frame keeps the molecules stable at a single-position with much less molecular motions, and thus enables robust single-molecule level TERS experiments at room temperature. Through this approach, we obtain TERS maps of single-level brilliant cresyl blue (BCB) molecules at room temperature, allowing us to probe the spatial heterogeneity of the single BCB molecules adsorbed on the Au surface. Furthermore, through the quantitative analysis of the measured TERS frequency variation through density functional theory (DFT) calculations, we provide a comprehensive picture of the conformational heterogeneity of single molecules at room temperature.

## Results and discussion

**Pre-characterization for ideal TERS conditions.** For highly sensitive single-molecule level detection at room temperature, we use bottom-illumination mode TERS, as illustrated in Fig. 1a. As a sample system, BCB molecules were spin-coated on a thin metal film and covered by an $Al_2O_3$ capping layer to suppress rotational and spectral diffusion[5]. The capping layer not only provides a freeze-frame for individual molecules, but also protects them from unwanted chemical contamination under ambient conditions, especially by sulfur[43] and carbon[44] molecules. Moreover, it prevents possible contamination of the Au tip, e.g., adsorption of the probing molecules onto the tip surface that can cause artifact signals, as shown in Fig. 1a (see Supplementary Fig. 1 for more details). It should be also noted that a previous study

demonstrated that photobleaching of BCB molecules is significantly reduced under vacuum environment compared to the ambient condition because oxygen in air causes the photo-decomposition process[30]. Hence, the $Al_2O_3$ capping layer is beneficial to reduce the photobleaching effect of molecules in our experiment. We used an electrochemically etched Au tip attached to a tuning fork for normal-mode atomic force microscopy (AFM) operation (see Methods for details). Using an oil-immersion lens (NA = 1.30), we obtained a focused excitation beam with a sub-wavelength scale, which can highly reduce the background noise of far-field signals in TERS measurements.

Furthermore, in combination with the radially polarized excitation beam, we achieved strong field localization in the normal direction with respect to the sample surface, i.e., a strong out-of-plane excitation field in parallel with the tip axis[45]. The excitation field, with a wavelength of 632.8 nm, is localized at the nanoscale tip apex, and the induced plasmon response gives rise to the resonance Raman scattering effect with the BCB molecules[41]. Figure 1b shows the far-field and TERS spectra of BCB molecules measured with linearly and radially polarized excitation beams (see Supplementary Fig. 2 for the plasmon response). With the exposure time of 0.5 s, we hardly observed the far-field Raman response of molecules (black), due to the extremely low Raman scattering cross-section. By contrast, we observed a few distinct Raman modes via the TERS measurements with a linearly polarized excitation (blue). Moreover, through the radially polarized excitation[46,47], we observed most of the normal modes with a substantially larger TERS intensity (red) compared to the TERS spectra measured with the linearly polarized excitation. The radially-polarized beam has much larger vertical field component after passing through a high NA objective lens compared to the linearly-polarized beam[48]. For example, the C-H₂ scissoring mode at ~1360 cm⁻¹ is clearly identified in the TERS spectrum measured with the radially polarized light, whereas it is not present in the TERS spectrum measured with the linearly polarized light.

**Optimization of the metal substrate for TERS.** In bottom-illumination mode TERS, the deposition of flat thin metal films on the coverslip is required to preserve the transparency of the substrate and to avoid SERS and fluorescence signals originating from the metal nano-structures[49]. To demonstrate the influence of the surface condition of metal films, we performed a control experiment based on Au films fabricated by four different conditions with two control parameters of the cleaning method and the deposition rate (see Supplementary Table 1 for detailed control parameters). From the AFM results of Fig. 1c–e, we verify that the optimal process is required for fabrication of flat metal thin films (see Supplementary Fig. 3 for detailed experiment results).

Another important parameter for bottom-illumination mode TERS is the metal film thickness, because a sufficiently thick metal film is required to induce strong dipole-dipole interactions between the tip dipole and the mirror dipole of the metal film[50]. However, the light transmission decreases with increasing metal thickness, which gives rise to a reduced excitation rate and collection efficiency in TERS. To experimentally determine the optimal thickness, we deposited Au films on $O_2$-plasma-cleaned coverslips with a Cr adhesion layer. We prepared six metal films with various thicknesses of 5, 7, 9, 11, 13, and 15 nm. Among these metal substrates, we could not perform TERS experiments with the 15 nm metal film because it was difficult to align the tip apex to the laser focus due to low light transmission. Regarding the 5 and 7 nm metal films, we could barely observe TERS signals from the BCB film because the TERS enhancement factor was too

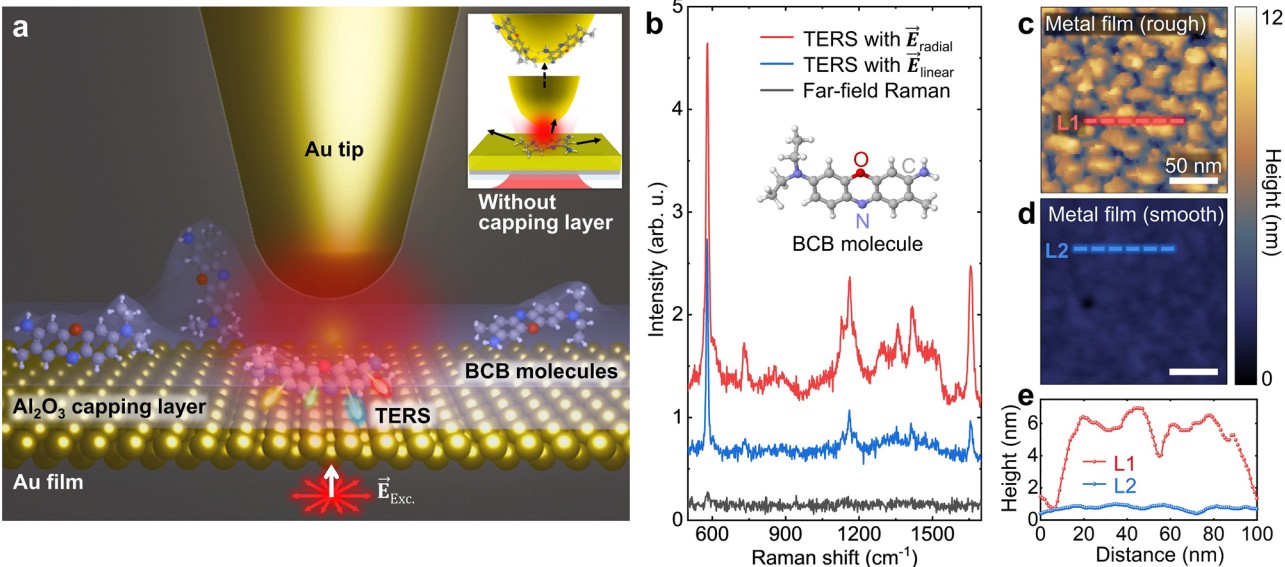

**Fig. 1 Schematic and pre-characterizations of bottom-illumination mode TERS. a** Schematic illustration of bottom-illumination mode TERS. The encapsulated BCB molecules on the Au surface are excited by a radially polarized beam ($\vec{E}_{Exc.}$) and the back-scattered TERS responses are collected. The inset illustration shows a tip-contamination process in conventional TERS without using an $Al_2O_3$ capping layer. **b** Far-field (black) and TERS spectra with different excitation polarization conditions. The TERS response with the radially polarized excitation laser (red) gives a larger enhancement compared to that obtained with the linearly polarized laser (blue). AFM topography images of thin metal films fabricated with different deposition rates and cleaning methods of the substrate (coverslip). The coverslip for the rough metal film (**c**) is cleaned using piranha solution, and the metal film is fabricated with a deposition rate of 0.01 nm/s. By contrast, the coverslip for the smooth metal film (**d**) is cleaned by ultrasonication in acetone and isopropyl alcohol along with $O_2$ plasma, and the metal film is fabricated with a deposition rate of 0.1 nm/s. **e** Topographic line profiles of two thin metal films, derived from **c** and **d**. Source data are provided as a Source Data file.

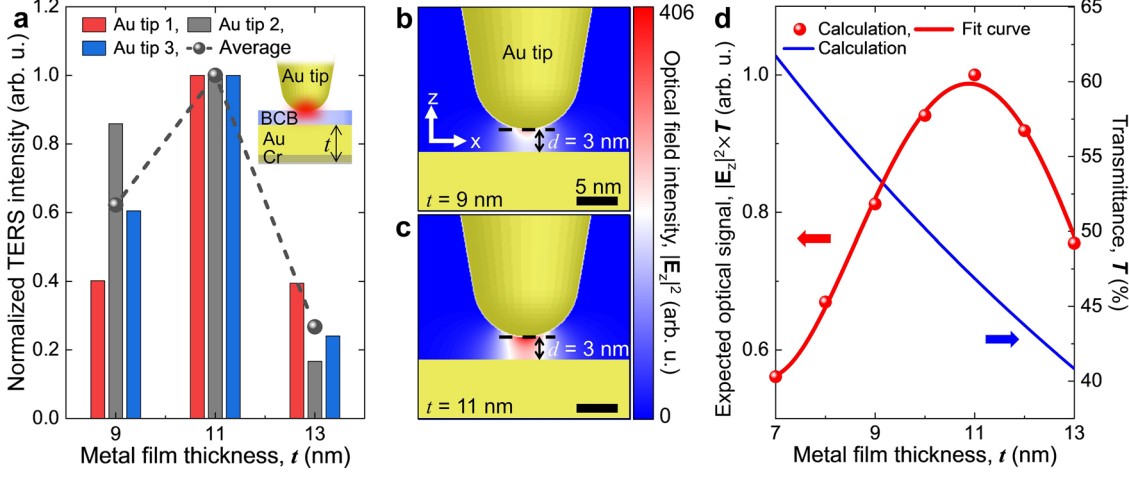

**Fig. 2 TERS enhancement comparison for different metal film thickness. a** Comparison of TERS intensity of BCB molecules for the metal substrates with different thicknesses ($t$ = 9, 11, and 13 nm). The ratio of tip-enhanced and tip-retracted Raman intensities of the ~580 $cm^{-1}$ peak is measured for different substrates with three different tips. The obtained intensity ratios for the three different tips are normalized to [0, 1] for comparison of the thickness effect. The black circles show average values for three tips. FDTD-simulated optical field intensity ($|\mathbf{E}_z|^2$) distribution at the nano-gap between the Au tip and the thin metal film with a film thickness of 9 nm (**b**) and 11 nm (**c**). **d** Expected optical signal (red circles) in the bottom-illumination mode and theoretically calculated transmittance (blue) with respect to the thickness of the metal film. The expected optical signal is calculated by the FDTD-simulated optical field intensity at the nano-gap multiplied by transmittance ($T$) at each film thickness. The derived result of the expected optical signal as a function of the metal film thickness is fit with a 4th degree polynomial (red line). $T$ is calculated with the equation (1). Source data are provided as a Source Data file.

low. Therefore, we performed a control experiment with three different metal substrates, namely with metal thicknesses of $t$ = 9, 11, and 13 nm.

To compare the relative TERS intensities of BCB film for these three metal substrates, we obtained the TERS spectra for these substrates with three different Au tips, i.e., each sample was measured using three Au tips. Figure 2a shows a comparison of

the measured TERS intensities with respect to the thickness of the metal films. We consider the strongest TERS peak at ~580 $cm^{-1}$ and determine the relative TERS intensities for different metal films. When we used three different tips for this control experiment, the TERS enhancement factors in each case were different; nevertheless, the metal film with 11 nm thickness yielded the strongest TERS signal for all the tips. Therefore, we

normalize the TERS intensity measured for the 11 nm metal film to [0, 1] for all three tips and compare the relative TERS intensities measured for the 9 and 13 nm metal films for each tip, as displayed in Fig. 2a. The black circles indicate the average TERS intensities for the three tips, for each substrate. The TERS intensities of the ~580 cm$^{-1}$ peak, measured for the 9 nm and 13 nm thick metal films, are ~30% and ~60% lower than that measured for the 11 nm metal film.

We then verified the ideal metal film thickness through theoretical approaches. First, we calculated the localized optical field intensity between the Au tip apex and the Au surface with respect to the metal film thickness using finite-difference time-domain (FDTD) simulations under the excitation light source ($\lambda = 632.8$ nm) placed below the Au film (see Methods and Supplementary Fig. 4 for details). Figure 2b and c show the simulated $\left|E_z\right|^2$ distributions for the metal film thicknesses of 9 nm and 11 nm, respectively (see Supplementary Fig. 5 for $\left|E_x\right|^2$ and $\left|E_{total}\right|^2$ components). When we set the distance $d$ between the Au tip and Au surface to $d = 3$ nm (i.e., the expected gap in tuning fork-based AFM), we achieve the maximum excitation rate for TERS, $\left|E_z\right|^2 \approx 400$, with the metal thickness of 11 nm. We subsequently calculate the expected optical signal, $\left|E_z\right|^2 \times T$, where $T$ is the calculated transmittance at $\lambda \sim 657$ nm by considering the strongest Raman peak of BCB at ~580 cm$^{-1}$ (see Supplementary Fig. 6 for details). Figure 2d shows the calculated signal as a function of the metal film thickness in the bottom-illumination geometry. $T$ is calculated with the following formula[51]:

$$T = \left|\frac{E(t)}{E_0}\right|^2 = e^{-4\pi\kappa t/\lambda},\qquad(1)$$

where $E_0$ and $E(t)$ are incident and transmitted optical field amplitudes, $\kappa$ is the extinction coefficient of Au at the given wavelength $\lambda$, and $t$ is the thickness of the metal film (see also Supplementary Fig. 7 for the experimentally measured transmittance)[52]. $\left|E_z\right|^2$ at each film thickness is obtained from FDTD simulations and multiplied by $T$, as the light passes through the metal film. $\left|E_z\right|^2 \times T$ is gradually enhanced with an increase in thickness up to $t = 11$ nm, but interestingly, it starts to decrease from 12 nm. To understand this behavior, we performed the same thickness-dependence simulations for different gaps between the tip and the metal surface (see Supplementary Fig. 8 for simulated results). Through these simulations, we found that the optimal metal film thickness varies slightly depending on the gap; nevertheless, the optimal metal film thickness is ~11–12 nm irrespective of the tip-surface gap (see also Supplementary Fig. 9 for the effect of Al$_2$O$_3$ at the tip-sample junction). Note that a previous study demonstrated that the plasmon resonance from a ~12 nm thick Au film gives rise to the largest TERS enhancement for the optical responses at ~640 nm[53]. Through the optimization process, the estimated TERS enhancement factor in our experiment is ~$2.0 \times 10^5$ (see Supplementary Note 1 for details), which is sufficient for single-molecule level Raman scattering detection as discussed in the previous study[54].

**Single-molecule level TERS imaging at room temperature**. We then performed the hyperspectral TERS imaging of single isolated BCB molecules adsorbed on the optimal metal film ($t = 11$ nm). First of all, we prepared a low-molecular density sample as described in Methods. Then, the freeze-frame (0.5 nm thick Al$_2$O$_3$) allowed us to stably detect single-molecule responses at room temperature (see Methods for details). Figure 3a and b shows the TERS integrated intensity images of the vibrational modes at ~580 cm$^{-1}$ (in-plane stretching mode of C and O atoms

in the middle of the molecule) and ~1160 cm$^{-1}$ (in-plane asymmetric stretching mode of O atom), which are only two recognizable TERS peaks of a single or a few BCB molecules, due to the short acquisition time (0.5 s) in our TERS mapping[30,41] (see also Supplementary Fig. 10 for the raw images of Fig. 3a and b). In the TERS images of both the ~580 cm$^{-1}$ and the ~1160 cm$^{-1}$ modes, the TERS intensity of the detected regions shows a spatial variation even though the responses are detected in similar nanoscale areas. This spatially heterogeneous intensity distribution originates from the difference in the number of probing molecules and/or the molecular orientation on the Au surface.

Since the apex size of the electrochemically etched Au tip is larger than ~15 nm, several molecules under the tip can be detected together, which gives rise to a strong TERS response. Alternatively, although some of the observed TERS responses are from single molecules, the Raman scattering cross-section can differ from molecule to molecule due to their orientations and the corresponding TERS selection rule[5]. Specifically, because the excitation field in our TERS setup has a strong out-of-plane polarization component, the peak-to-peak Raman scattering intensity changes depending on the conformation of a molecule. Hence, the conformational heterogeneity of probed molecules can be best exemplified by the peak intensity ratio of the vibrational modes at ~580 cm$^{-1}$ (Fig. 3a) and at ~1160 cm$^{-1}$ (Fig. 3b), as shown in Fig. 3c. Since TERS tip selectively probes the out-of-plane modes with respect to the surface, the peak intensity ratio can be low for multiple molecules in comparison with the single molecules. It should be noted that no structural evidence of single molecules was found in the simultaneously measured AFM topography image (Supplementary Fig. 11). Figure 3d shows time-series TERS spectra at a single spot exhibiting robust signals without spectral fluctuations owing to the freeze-frame effect. From this result, we expect stationary conformation of the molecules in the measurement area of Fig. 3a–c. By contrast, fluctuating TERS spectra with respect to time are observed without the capping layer (see Supplementary Fig. 12 for comparison).

From the TERS response corresponding to the nanoscale regions in the TERS image, we can infer the possibility of single-molecule detection; nevertheless, more substantial evidence is needed to verify this possibility. In addition to the aforementioned molecular orientation and the selection rule, the vibrational energy of the normal modes of an adsorbed molecule can change due to coupling with the atoms of the metal surface, leading to peak shift and intensity change signatures[4,55,56]. Additionally, the peak linewidth should be considered to distinguish the molecular ensembles from the single or a few molecules[29]. Further, to distinguish the single, a few, or multiple molecules, we also need to consider the spatial distribution in TERS images in addition to spectroscopic information[57]. Hence, we analyze the spectral properties and spatial distribution of the observed spots in the TERS image to obtain the evidence of single-molecule detection. We classify the observed TERS spots in Fig. 3 into two groups, as shown in Fig. 4a. We surmise that the TERS response in the first group (red circled spots 1–3) was measured from multiple BCB molecules because the TERS signal of both the ~580 cm$^{-1}$ and the ~1160 cm$^{-1}$ modes is pronounced, significant TERS peak shift is not observed, and the peaks have generally broad linewidths, as shown in Fig. 4b and c (see also Fig. 3a and b). By contrast, we observe much weaker TERS responses in the second group (blue circled spots 4–9 in Fig. 4a) with a significant peak variation corresponding to ~580 cm$^{-1}$, as large as ~7.5 cm$^{-1}$, as shown in Fig. 4d (see Supplementary Fig. 13 for the full spectral range). We then consider the linewidth of each spectrum and spatial distribution in the TERS image to distinguish the single and a few molecules. First of all, we classify the spot 4 as a few molecules.

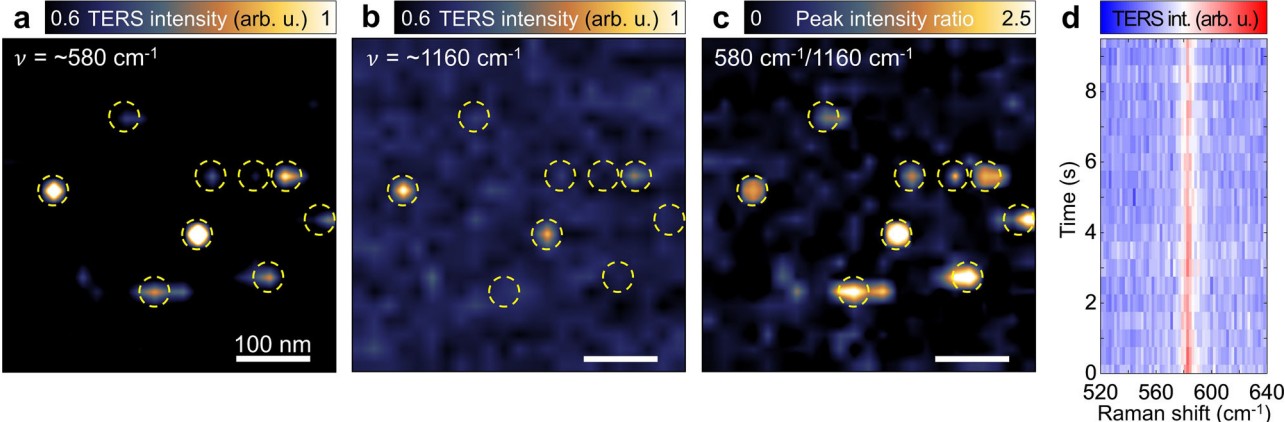

**Fig. 3 Single-molecule level TERS images and effect of the Al$_2$O$_3$ capping layer.** TERS mapping images of single BCB molecules measured with the excitation laser power of 220 μW and acquisition time of 0.5 s at each pixel at room temperature. TERS peak intensity images for in-plane symmetric stretching mode of O-C$_2$ and N-C$_2$ observed at ~580 cm$^{-1}$ (**a**) and in-plane asymmetric stretching mode of O-C$_2$ observed at ~1160 cm$^{-1}$ (**b**). **c** TERS peak-to-peak intensity ratio image of ~580 cm$^{-1}$ and ~1160 cm$^{-1}$ peaks, arithmetically calculated from TERS images **a** and **b** after subtracting a background fluorescence signal. Yellow dashed circles in **a–c** indicate the same positions in the TERS images. **d** Time-series TERS spectra at a single fixed position. Source data are provided as a Source Data file.

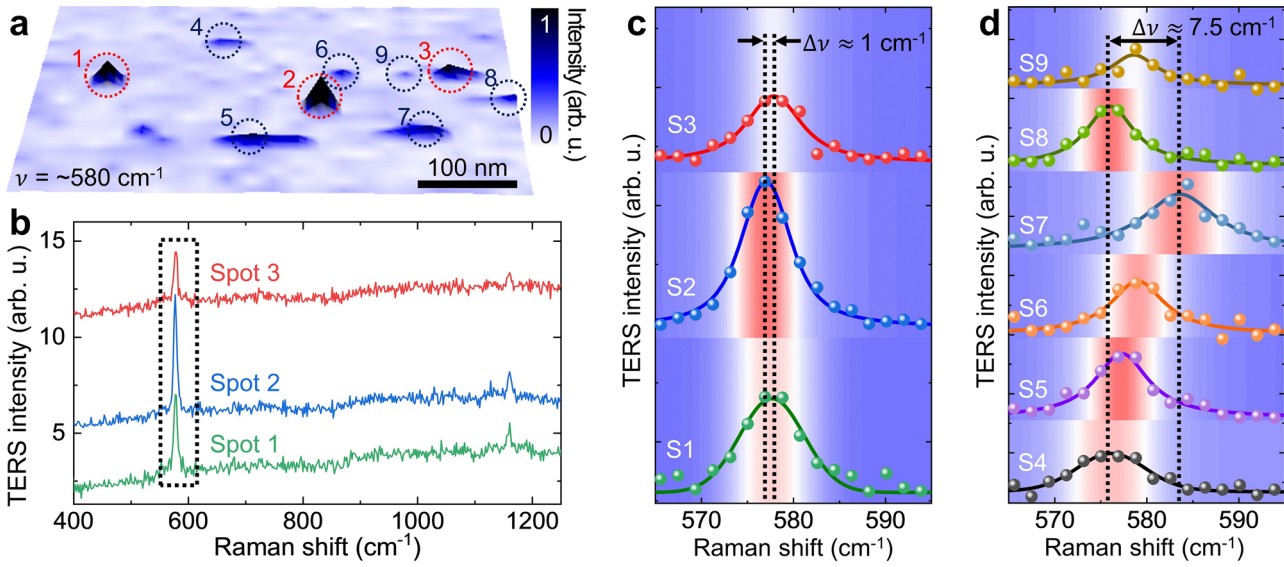

**Fig. 4 TERS spectra of each isolated BCB molecules. a** TERS peak intensity image of the vibrational mode at ~580 cm$^{-1}$ of BCB molecules. **b** TERS spectra measured at spots 1, 2, and 3 indicated with red circles in **a**. TERS spectra measured at spots 1–3 (**c**) and spots 4–9 (**d**) fitted with the Voigt line shape function in the range from 565 cm$^{-1}$ to 595 cm$^{-1}$. From the observed TERS peak shift in **d**, the TERS spectra at spots 1–3 (red circles in **a**) are probably measured from molecular ensembles and spots 4–9 (dark blue circles in **a**) are possibly measured from single or a few molecules. In **c** and **d**, the dots and lines are experimental data and fitted curves, and the TERS spectra at each spot are also shown as background 2D contour images. Source data are provided as a Source Data file.

Although its TERS intensity is weak with the small spatial distribution, the linewidth is quite broad (7.5 cm$^{-1}$) comparable to the first group (see Supplementary Table 2 for details). It should be noted that a few molecules can show broad linewidth even with double peaks when the molecules have different conformations (see Supplementary Fig. 14 for details). The remaining five spots (S5-9) show narrow linewidths of ≤6.0 cm$^{-1}$, which is closer to the spectral resolution in our experiment (see also Supplementary Fig. 12). Hence, we finally classify these spots into single or a few molecules group based on the spatial distribution in TERS image. As can be seen in Fig. 4a, the TERS response of spots S5, S7, and S8 is spatially spread out, which means the signals were obtained from a few molecules. Therefore, from the spatio-spectral analyses, we believe the TERS responses from S6 and S9 are possibly originated from single isolated BCB molecules.

**DFT calculation of vibrational modes in different chemical environments.** To reveal the possible origins of the observed TERS peak variations, we calculated the normal vibrational modes of a BCB molecule through DFT simulations. Since the BCB molecules are encapsulated using a thin dielectric layer, we presume the spectral diffusion is suppressed, as experimentally demonstrated in Fig. 3d. Based on this assumption, we design two kinds of fixed conformations of a BCB molecule, i.e., horizontal and vertical geometries with respect to the Au (111) surface. Regarding horizontally laying molecules, we additionally consider the position of the BCB molecules (especially C atoms vibrating with a large amplitude for the ~580 cm$^{-1}$ mode) with respect to the Au atoms since the substrate-molecule coupling effect can be slightly changed (see Methods for calculation details).

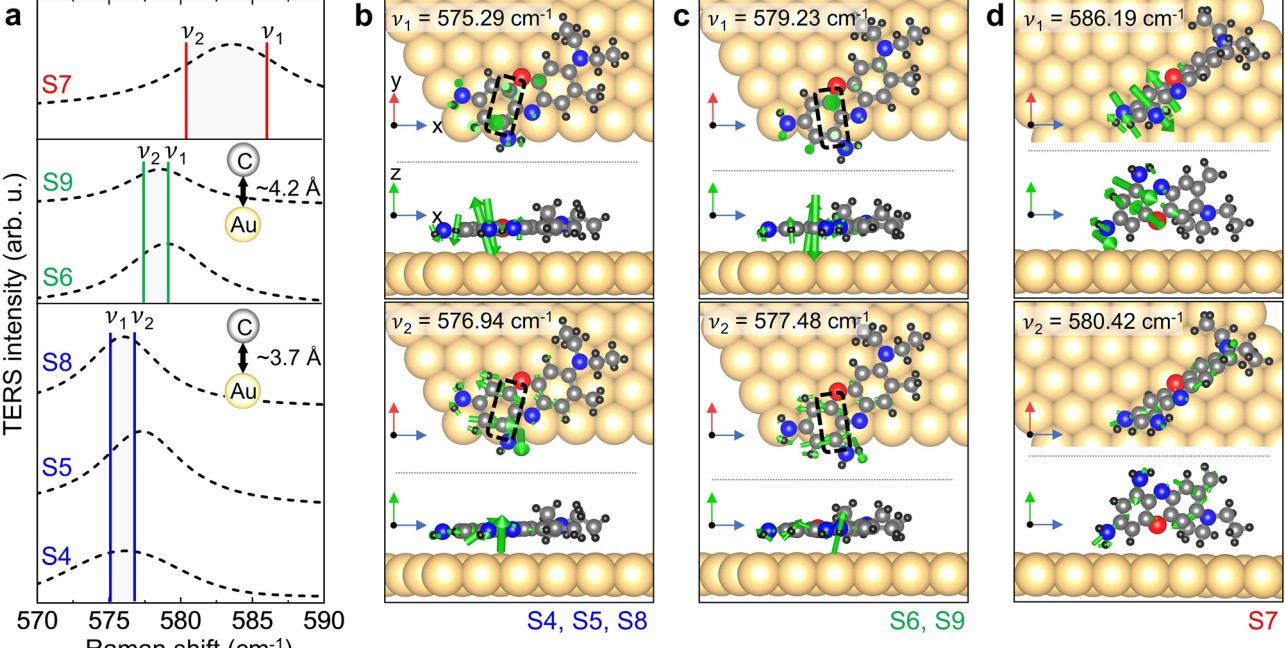

**Fig. 5 DFT-calculated vibrational modes of the BCB molecules. a** The measured TERS spectra (dashed curves) derived from Fig. 4d and DFT-calculated normal modes (colored vertical lines) for different chemical environments and molecular orientations. S4–S9 are the same signals as spots 4–9 in Fig. 4. **b–d** Models of a BCB molecule on the Au surface for the DFT calculations of normal modes in different conditions exhibiting the origin of the observed peak shift of single-molecule TERS measurements. The direction and size of the green arrows indicate the motion of each atom and its intensity. Source data are provided as a Source Data file.

Figure 5a shows the calculated normal modes (colored vertical lines) of a BCB molecule with the measured TERS spectra at spots 4–9 (Fig. 4a and d) for different chemical environments described in Fig. 5b–d. In the frequency range of 570–590 cm$^{-1}$, two theoretical vibrational modes ($v_1$ and $v_2$) are observed even though only a single peak was experimentally observed due to the limited spectral resolution and inhomogeneous broadening at room temperature. As individual atoms in a BCB molecule involve additional coupling to Au atoms on the surface, the Raman frequencies of two vibrational modes are varied depending on the conformation and the position of the molecule. When the two strongly oscillating C atoms of the molecule (as indicated with black dashed rectangles in Fig. 5b) are closer to the nearest Au atoms (the average atomic distance of two carbon atoms with the Au atom is 3.7 Å), $v_1$ is calculated as 575.29 cm$^{-1}$ with the out-of-plane bending vibration mode of the C atoms and $v_2$ is calculated as 576.94 cm$^{-1}$ with the in-plane stretching mode of the C atoms. By contrast, when these C atoms (as indicated with black dashed rectangles in Fig. 5c) are vertically mis-located with longer atomic distance with respect to the Au atoms (the average atomic distance of two carbon atoms with the Au atom is 4.2 Å), the Raman frequency of the out-of-plane bending mode of the C atoms is increased to 579.23 cm$^{-1}$ ($v_1$ in Fig. 5c). It is likely that the two strongly oscillating C atoms having shorter atomic distance with the Au atoms (Fig. 5b) experience stronger damping forces than the C atoms located further away from the closest Au atoms (Fig. 5c). On the other hand, both $v_1$ and $v_2$ are significantly increased for a vertically standing molecule (Fig. 5d) due to the lessened molecular coupling with the Au atoms (see Supplementary Fig. 15 for the normal mode of a BCB molecule in the gas phase).

From these simulation results, we can deduce that the experimentally observed possible single molecules in S6, and S9 (in Figs. 3 and 4) likely have similar molecular orientations to the illustrations in Fig. 5c. The other chemical or environmental conditions give much less effect to the spectral shift compared to the molecular orientation and coupling (mainly C-Au atoms).

The observed molecule in S8 is expected to have same molecular orientation as the molecules in S4, and S5 with the C atoms vertically aligned with respect to the Au atoms, as shown in Fig. 5b. The observed broader linewidth and higher frequency TERS peak at S7 indicate the molecule in S7 is oriented vertically, as displayed in Fig. 5d. Hence, in this work, we experimentally verified the freeze-frame effect using a thin dielectric layer and probed the conformational heterogeneity of possible single molecules at room temperature through highly sensitive TERS imaging and spectral analyses with DFT simulations.

We exclude other possible effects of the observed spatial shifts for the following reasons. First, spectral shifts can occur depending upon the chemically distinct states, i.e., protonation and deprotonation, of molecules due to the local pH differences[58]. However, in our work, the spin-coated BCB molecules are physisorbed on the Au surface (not chemically bound) and the proton transfer process rarely occurs because of the encapsulating dielectric layer on the molecules[59]. Second, spectral shifts can be observed by the hot-carrier injection from the plasmonic metal to molecules because it can cause a change in molecular bond lengths[60]. However, in our experiment, we exclude this hot-carrier injection effect because the tip-molecule distance is maintained at ~3 nm and the dielectric capping layer also suppresses the hot-carrier injection. Third, a previous study demonstrated that oxidation-reduction reaction of molecules could be induced by electrochemical TERS[61]. However, since we do not apply external bias to the BCB molecules, we believe the redox effect is negligible in our observed spectral shifts. Lastly, the Stark effect was experimentally observed for single molecules sandwiched with the metal nano-gap when the applied DC electric fields at the gap was large enough (~10 V/nm)[62]. Since we do not apply external bias to the BCB molecules, we can ignore the Stark effect.

In summary, we demonstrated the hyperspectral TERS imaging of possibly single molecules at room temperature by optimizing experimental conditions. In addition, the thin dielectric $Al_2O_3$ layer encapsulating the single molecules adsorbed onto the Au (111) surface played a significant role, as a freeze-frame, in enabling room temperature single-molecule TERS imaging. This is because the thin dielectric layer can suppress the rotational and spectral diffusions of molecules and inhibit the chemical reactions and contaminations in air, including potential physisorption of molecules onto the Au tip[63,64]. Through this room-temperature TERS imaging approach at the single-molecule level, we examined the conformational heterogeneity of BCB molecules with supporting theoretical DFT calculations. We envision that the presented optimal experimental setup for single-molecule TERS measurements will be broadly exploited to investigate unrevealed single-molecule characteristics at room temperature. For example, we can investigate intramolecular vibrational relaxation more accurately at the single-molecule level using this freeze-frame and variable-temperature TERS[5]. In addition, the single-molecule strong coupling study at room temperature will be more easily accessible and various advanced studies will be enabled[6], such as tip-enhanced plasmon-phonon strong coupling and investigation of the coupling strength with respect to the molecular orientation. Furthermore, this approach can extend to the single-molecule transistor studies at room temperature with very robust conditions, e.g., suppressing spectral fluctuations, photobleaching, and contaminations.

## Methods

**Sample preparation**. Coverslips (thickness: 170 μm) for non-optimal rough metal films (Fig. 1c) were cleaned with piranha solution (3:1 mixture of $H_2SO_4$ and $H_2O_2$) for >60 min and ultrasonicated in deionized water. The other coverslips for optimal smooth metal films (Fig. 1d) were ultrasonicated in acetone and iso-propanol for 10 mins each, followed by $O_2$ plasma treatment for 10 mins. The coverslips were then deposited with a Cr adhesion layer with a thickness of 2 nm (a rate of 0.01 nm/s) and subsequently deposited with Au films with varying thick-nesses (0.01 nm/s to 0.1 nm/s) at the base pressure of $\sim 10^{-6}$ torr using a conventional thermal evaporator. The deposition rate of metals was precisely controlled using a quartz crystal microbalance detector. The transmittance spectra of the substrates were measured by a UV–Vis spectrometer (UV-1800, Shimadzu). Next, BCB molecules in ethanol solution (100 nM for the single-molecule experiment and 1.0 mM for the thickness optimization process) were spin-coated on the metal thin films at $150 \times g$ (3000 rpm). Finally, an $Al_2O_3$ capping layer with a thickness of 0.5 nm was deposited on the sample surface using an atomic layer deposition system (Lucida D100, NCD Co.). The $Al_2O_3$ layer was deposited with the growth rate of 0.11 nm per cycle at the temperature of 150 °C under the base pressure of 40 mTorr. Precursor for the $Al_2O_3$ ALD was trimethylaluminum (TMA) vapor, and $H_2O$ vapor as the oxidant with $N_2$ carrier gas.

**TERS imaging setup**. For TERS experiments, we used a commercial optical spectroscopy system combined with an AFM (NTEGRA Spectra II, NT-MDT). The excitation beam from a single-mode-fiber-coupled He-Ne laser ($\lambda = 632.8$ nm, optical power $P$ of $\geq 100$ μW) passed through a half-wave plate ($\lambda/2$) and was collimated using two motorized lenses. The beam passed through a radial polarizer and was focused onto the sample surface with an oil immersion lens (NA = 1.3, RMS100X-PFOD, Olympus) in the inverted optical microscope geometry. The electrochemically etched Au tip (apex radius of $\sim 10$ nm) attached on a tuning fork was controlled using a PZT scanner to alter the position of the Au tip with respect to the focused laser beam. The backscattered signals were collected through the same optics and transmitted to a spectrometer and charge-coupled device (Newton, Andor) after passing through an edge filter (633 nm cut off). The spectrometer was calibrated using a mercury lamp and the Si Raman peak at 520 cm$^{-1}$, and its spectral resolution was $\sim 4.3$ cm$^{-1}$ for a 600 g/mm grating.

**FDTD simulation of optical field distribution**. We used FDTD simulations (Lumerical Solutions, Inc.) to quantify the optical field enhancement at the apex of the Au tip with respect to the metal film thickness. The distance between the Au tip and the Au film was set to 2–4 nm based on our experimental condition. As a fundamental excitation source, monochromatic 632.8 nm light was used with linear polarization parallel to the tip axis. The theoretical transmittance of thin gold films was calculated using the material properties obtained from ref. [52].

**DFT calculation of BCB vibrational modes**. DFT calculations were performed using the Vienna Ab initio Simulation Package to identify the normal vibrational modes of a single BCB molecule. In our simulation model, the chemical environment and molecular orientation were critically considered; i.e., the vibrational modes for a BCB molecule placed in a free space and adsorbed onto the Au (111) surface in different orientations and positions were calculated. Specifically, molecules oriented normal and parallel to the Au surface were modeled. For a molecule placed parallel to the Au surface, the lateral position was varied, as shown in Fig. 5, to consider the effect of an interaction between the atoms of the BCB molecule and those of the Au surface. To describe the adsorption of a molecule on the Au (111) surface, Tkatchenko and Scheffler dispersion correction[65] with the Perdew–Burke–Ernzerhof exchange functional[66] was employed. Regarding the Au (111) slab supercell, four layers of Au atoms were considered, with the top two layers optimized in the gamma point. Furthermore, 1.5 nm vacuum space was considered to avoid an interaction between periodic slabs, and the plane-wave energy cutoff was set to 550 eV. The vibrational modes of the BCB molecule were shown using visualization for electronic and structural analysis.

## Data availability

The data that support the findings of this study are available from the corresponding author upon reasonable request. Source data are provided with this paper.

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

## Acknowledgements

M.K., Y.K., H.L., and K.-D.P. acknowledge funding from the National Research Foundation of Korea (NRF) grants (No. 2019K2A9A1A06099937 and 2020R1C1C1011301) and Basic Science Research Institute Fund, whose NRF grant number is 2021R1A6A1A10042944. G.L. acknowledges the computing resources supported by the UNIST Supercomputing Center. Y.D.S. acknowledges funding from the Global Research Laboratory (GRL) Program through NRF funded by the Ministry of Science and ICT (No. 2016911815), grant (KK2061-23 and KK2161-22) from the Korea Research Institute of Chemical Technology, the 2022 Research Fund (1.220108.01) of UNIST(Ulsan National Institute of Science & Technology), and support from the Industrial Strategic Technology Development Program (No. 10077582) funded by the Ministry of Trade, Industry, and Energy (MOTIE), Korea. H.K. acknowledges grant (SI2131-50) from the Korea Research Institute of Chemical Technology.

## Author contributions

K.-D.P. and M.K. conceived the experiment. M.K. prepared the sample with the partial contribution of H.K. and T.E. M.K., Y.K., and H.L. performed the measurements. E.O., H.J., J.C., and G.L. performed the simulations. K.-D.P. and M.K. analyzed the data, and all authors discussed the results. M.K. and K.-D.P. wrote the manuscript with contributions from all authors. K.-D.P. and Y.D.S. supervised the project.

## Competing interests

The authors declare no competing interests.
