## [Peer Review File · Nature Communications]

Reviewer comments, first round review

Reviewer #1 (Remarks to the Author):

The authors Kang et al. use a thin Al₂O₃ coating to immobilize individual or few molecules of brilliant cresyl blue (BCB) on a metal surface. This allows them to study the molecules in ambient conditions as the Al₂O₃ coating "freezes" the molecules in place preventing diffusion and conformational changes while probing with tip enhances Raman spectroscopy. They tune the thickness of the gold layer and test three different electrochemically etched tips to optimise signal. Finally, the authors compared the observed single (or few) molecule signals and differences between them, with density functional theory computations of different conformations/orientation of the probed molecule and assign a particular conformation to the observed single molecule signals.

The work is thorough, well written and in particular the immobilization of analytes for room temperature TERS measurements is very interesting. However, I do have some concerns.

Mainly, very little discussion is offered regarding alternative explanations for the observed shifts in vibrational energies. I would like to see whether the authors have considered alternative hypotheses and how these were eliminated.

For example, spectral shifts can also occur upon protonation/deprotonation as local pH differences on such small length-scales are a possibility (see e.g. 10.1126/sciadv.abg1790).

The same is true for charging or redox effects (see e.g. 10.1021/acs.jpcc.9b12002), and the Stark effect (see e.g. 10.1021/acs.jpcclett.8b01343)

Also, have the authors investigated and eliminated whether any interaction between the BCB and the Al₂O₃ can cause these particular shifts?

While I understand that not all alternative explanations can be fully modelled with DFT and compared, the authors should make a decent effort in considering alternative explanations and include a robust discussion on the subject, supported by either data, modelling or reasoning/justification for elimination.

Can the authors provide the entire TERS spectrum retrieved for points 5-9? Maybe they can be included in the SI.

In the introduction, the authors oversell the TERS technique by stating that: "SERS based approaches do not allow single-molecule measurements in heterogeneous chemical environments..." It is not clear what is meant by the heterogeneous environment in this manuscript as only one analyte molecule is probed but in a matrix of Al₂O₃, but in such an environment measuring single molecules using SERS is certainly possible (see e.g. 10.1038/s41467-021-26898-1).

What step size was used when collecting the TERS images, and has any smoothing, averaging or interpolation been used in the images Fig. 3 a,b and 4a? If so then this should be reported, and raw images shown in the SI.

Towards the end the authors use the term: "...likewise illustrations in Fig..." This is incorrect and should be revised. The term chosen to replace should be chosen carefully and justified with the aforementioned discussion, as it states the strength/confidence with which the authors claim the geometry in Fig. 5c is the geometry observed in the experiments (based on the match to the Raman spectrum).

Reviewer #2 (Remarks to the Author):

This manuscript reports on tip enhanced hyperspectral Raman imaging of brilliant cresyl blue (BCB) at the single molecule level at room temperature. The carefully prepared samples consisted

of thin, smooth Au-films deposited on low optical background covers slides. A low concentration of BCB molecules deposited on the gold film by spin coating. The sample was covered with a protective capping layer of 0.5 nm Al₂O₃ to suppress spatial and spectral diffusion of the molecules. The optical measurements were performed with an inverted confocal microscope by illuminating the BCB-molecules with a radially polarized beam through the gold film to form a coupled plasmon oscillation between the apex of close sharp gold tip and the gold film right below. By blocking the incident radiation from reaching the detector, images were recorded by raster scanning the sample and detecting the back scattered light from the sample point by point. Due to the low concentration of the BCB molecules on the gold film dim and bright spots with diameters of about 20 nm could be observed in the Stokes shifted part of the spectrum. From these spots Raman spectral of the BCB where recorded. The Raman spectra of nine isolated spots where analyzed. The line widths of the most intense Raman peaks located around 580 cm⁻¹ varied from 7.5 cm⁻¹ to 5.4 cm⁻¹ and their center frequencies differed by as much as 7.5 cm⁻¹. This inhomogeneous behavior indicates that the individual spots may originate from single isolated molecules. Abrupt spectral fluctuation typically observed in single molecule spectroscopy where not observed for the samples with an applied protective layer, however they were observed from unprotected samples and were associated with a considerable spectral line broadening.

Single-molecule TERS is challenging at ambient conditions due to the rapid structural dynamics of the surface bound molecules exposed to the atmosphere. In recent years, TERS-measurements performed in UHV and at cryogenic temperatures in an STM-configuration have excited considerable sensation since they could be performed with very high spatial resolution revealing single atoms in a molecule and single chemical bonds. However, room temperature is required for a large number spectroscopic studies, e.g. to investigate molecular dynamics and functions relevant e.g. for life sciences, biology or energy conversion to name a few. This work is a significant contribution to single-molecule SERS or TERS and has several interesting aspects. 1) It introduces a clever way to reduce fast dynamics and considerably reduces spectral diffusion or spectral broadening leading to highly resolved Raman spectra at ambient temperatures making it interesting for a wide audience. 2) Raman scattering is performed by exciting through a thin Au-film a coupled plasmon oscillation between the tip and the sample where the vibrational modes of the molecule can perfectly modulate the plasmon oscillation. Hence a strong polarization sensitive Raman spectrum of a single molecule can be detected while at the same time the background of out of focus features are suppressed. This is an interesting and highly welcome alternative to the often used side-illumination scheme. Furthermore, it has to be mentioned that the authors have optimized the thickness of their Au-films to obtain a strong back-scattering signal from the coupled gap-plasmon mode. 3) Extensive quantum chemical calculations have been performed to explain the heterogeneity in the Raman spectra in terms of the orientation and localization of the molecules with respect to different positions on the Au-film.

This work provides an easy way to investigate the single-molecule properties in interacting molecules at room temperature and hence expands the scope of single-molecule vibrational spectroscopy studies.

The experimental approach, the quality of the data and their analysis are valid and the contents is clearly and scholarly presented. I suggest publication after some minor but mandatory revisions that should help to further improve this manuscript.

Questions and suggestions to be consider in the review:

- What effect has the Al₂O₃ film on the plasmonic gap field distribution between the tip apex and the gold film?
- Does the capping layer also protect the BCB-molecules against photo-bleaching?
- Can the authors comment on the effect of the positive charge of the BCB-molecules on the Au-layer and the gap field?
- Often TERS spectra of Au-samples probed with an Au tip show a significant background due to interband transitions and inelastic plasmon decay. Why don't we see this background in Fig 1b) ?

- The radial laser mode gives a dominant longitudinal field component in the center which is wanted to excite a strong gap plasmon-polariton field and allows for polarization sensitive measurements. My question points to the azimuthally polarized ring which surrounds the optical axis. For a 1.3 NA objective lens this field intensity component is still significant and increases the focal diameter and probably contributes to the PL background from Au. On the other hand it can efficiently be scattered from horizontal molecule revealing in-plane vibrational modes. Can the authors comment on this issue and say why they have not used a higher NA? Hence, it would be very worthwhile to also add the calculated radial field components either along with Fig. 2b and Fig. 2c or add them to the SI.

- A gap mode created between a smooth gold sample and a sharp gold tip excited by radial polarization has been used for tip enhanced Raman and luminescence spectroscopy before. An article closely related to this work that should be cited is "Imaging nanometre-sized hot spots on smooth Au films with high-resolution tip-enhanced luminescence and Raman near-field optical microscopy" by M Sackrow et al, *ChemPhysChem* 9 (2), 316-320, 2008.

- I do not agree with the following statement: "However, SERS-based approaches do not allow single-molecule measurements in heterogeneous chemical environments due to their diffraction-limited spatial resolution." Single-molecule measurements based on SERS in heterogeneous chemical environments have been shown and investigated in detail by several authors, exhibiting inhomogeneous dynamic spectral behavior due to the heterogeneous nature e.g. of the Ag-aggregate, although diffraction limited. The authors should formulate their statement more clearly and should also cite some of the pioneering single-molecule SERS references, which show such spectral fluctuations e.g. S. Nie and S. R. Emory, *Science* 275, 1102-1106 (1997). K. Kneipp, et al., *Phys. Rev. Lett.* 1997, 78, 1667- 167, T Vosgröne, AJ Meixner, *ChemPhysChem* 6 (1), 154-163, 2005. I recommend that the authors check out a very recent SERS- article by M Pszona et al. *J Chem Phys* 156, 014201, 2022.

Reviewer #3 (Remarks to the Author):

This work from the Park group is nicely presented with some beautifully performed experiments supported by numerical simulations. The authors have claimed to use the "freeze-frame" approach by trapping the sample molecules under a thin layer of aluminum oxide and have shown TERS images of BCB molecules. I have a few concerns, as mentioned below, and based on them, I suggest a minor revision of this paper before it can be accepted for publication in *Nature Communications*.

The authors claim to have imaged single molecules in this work. Although they slightly toned down the claim by adding the word "possibly" and say that they have demonstrated hyperspectral TERS imaging of "possibly single molecule" at room temperature. Although I highly rate this work, I am not satisfied with the claim of a single molecule, even with the toned-down claim. The authors need to provide more convincing evidence for possible single molecule observation.

First, TERS has two features, high spatial resolution and high field enhancement. Both or either can be used for single molecule detection. That means, either the spatial resolution should be extremely high to distinguish every molecule individually, or the sample should be prepared in such a diluted way that molecules are isolated and only one molecule is present in the observation volume. In this work, the authors have not said anything about the spatial resolution, but one can see from Figure 3a that it is about 20 nm, which is a typical value in TERS imaging. With such spatial resolution, the only possibility is to prepare an extremely diluted sample to make sure molecules are isolated, which seems to be the case here in this paper. Once such a sample is prepared, there is nothing much that TERS does, except locating the molecule and enhancing the signal to a level that a single molecule can be observed. Such enhancement factor should be at least a million times, if not more. Is the aim of the paper related to high enhancement in TERS? However, the authors have not discussed anything about the enhancement factor. As I mentioned, once a good sample is prepared, TERS does not have much role to play other than huge enhancement. So my first suggestion is that the authors should estimate the enhancement factor

and discuss if it is sufficient for the observation of a single molecule or not. Although, I do believe that the enhancement is sufficiently high, otherwise authors will not observe such a sample by TERS. Or, if the goal of this work is something other than what I have mentioned, then the authors should clearly mention that.

My second point is about how the authors distinguish between single and multiple molecules in Fig. 4, which is later supported by simulation in Fig. 5. The authors claim that spots 1-4 with small shifts in peak are multiple molecules while spots 5-9 with large shifts in peak are single molecules. This is not very clear to me. If there are multiple molecules, and if they have different conformations (for example, one molecule has conformation as in Fig. 5b and another as in Fig. 5d), one would effectively see a large peak shift depending upon the individual conformations. Also, there will be effectively large peak widths for the same reason. This would then be the case of Fig. 4d rather than Fig. 4c. Moreover, Figs. 3a, 3c and 4a clearly show that spots 5, 7 and 8 do not represent a single molecule because these points have large spread (up to 100 nm in size in the x-direction) in the images. Considering the spatial resolution to be around 20 nm, this is possible only if multiple molecules are aligned along the x-direction. Or is there something that I am missing here? The authors should clarify how they distinguish single and multiple molecules, because their current explanation is not very clear.

In addition to my concerns mentioned above, I have a few minor points-

1. Overall, the paper is written in a clear way, but there are some issues with the language here are there. The authors should pay more attention to the language.
2. The idea of trapping the sample molecules under a thin layer of aluminum oxide to restrain small fluctuation in position or orientation of sample molecules at room temperature is very nice. However, the authors have not mentioned if this capping also affects the molecular vibration or not? If it does, has it been taken into consideration? Although, even if the molecular vibrations are altered, it will not affect the results of this work.
3. The authors mention that the aluminum oxide layer on the sample molecule prevents "unnecessary contamination" of the tip. Although I understand what the authors mean and it is also explained through Fig. S1, it is not the usual way to refer to tip contamination in TERS. Since the experiment is performed in ambient conditions, there are many possibilities for the tip to get contaminated with foreign molecules. Carbon and sulfur are the most common contaminations during TERS experiments, which come from the atmosphere. Instead of simply using the word "contamination", the authors should instead clearly mention that they could prevent the sample molecules from getting picked up by the tip during the imaging process. Also, it is incorrect to say that the thin layer "protects plasmonic tip"! It rather protects the sample molecules.
4. When a metallic tip is used in combination with a metallic substrate, one needs to consider not only the transmission of light through the substrate, but also the possible hybridization of plasmons, which affects the enhancement at a given wavelength. The authors should refer to a previous paper (Nanoscale, vol. 4, p. 5931, 2012), where a similar result was shown for a similar wavelength. The reference paper demonstrated that for 642 nm, the best enhancement in TERS was obtained when the substrate was 12 nm thick.
4. The reference of Fig. S3 comes after that of Fig. S5 in the main text. Please take care of the sequence.
5. The intensity ratio in Fig. 3c is not clear. Should the ratio depend on the molecule number or the conformation of the molecule, or should it be uniform? What is the reason that some spots are much stronger than others? For example, spots 5, 7 and 8, which are supposed to be single molecules, have much more intensity than spots 1, 3, 4, which are supposed to be multiple molecules.

Reviewer #1 (Remarks to the Author):

The authors Kang et al. use a thin Al₂O₃ coating to immobilize individual or few molecules of brilliant cresyl blue (BCB) on a metal surface. This allows them to study the molecules in ambient conditions as the Al₂O₃ coating "freezes" the molecules in place preventing diffusion and conformational changes while probing with tip-enhanced Raman spectroscopy. They tune the thickness of the gold layer and test three different electrochemically etched tips to optimize signal. Finally, the authors compared the observed single (or few) molecule signals and differences between them, with density functional theory computations of different conformations/orientation of the probed molecule and assign a particular conformation to the observed single molecule signals.

The work is thorough, well written and in particular the immobilization of analytes for room temperature TERS measurements is very interesting. However, I do have some concerns.

We sincerely thank the reviewer for acknowledging the novelty and significance of our work. In addition, we appreciate the constructive comments to improve our manuscript. With regard to the raised concerns below, we have addressed them clearly and provided supporting results in the point-by-point response below. We have also made corresponding revisions to our manuscript as indicated in red at the advice of the reviewers.

Mainly, very little discussion is offered regarding alternative explanations for the observed shifts in vibrational energies. I would like to see whether the authors have considered alternative hypotheses and how these were eliminated. For example, spectral shifts can also occur upon protonation/deprotonation as local pH differences on such small length-scales are a possibility (see e.g. 10.1126/sciadv.abg1790). The same is true for charging or redox effects (see e.g. 10.1021/acs.jpcc.9b12002), and the Stark effect (see e.g. 10.1021/acs.jpcclett.8b01343)

We agree with the reviewer that we should provide more discussion on the other possibilities of the observed spectral shifts and the reasons why we could exclude those possibilities.

1. Regarding the protonation reactions, as the reviewer mentioned, spectral shifts can occur depending upon the chemically distinct states (protonation and deprotonation) of molecules due to the local pH differences [Huang et al., *Sci. Adv.* **7**, eabg1790 (2021)]. However, in our work, the spin-coated BCB molecules were physisorbed on the Au surface (not chemical binding) and the proton transfer process was rarely occurred because of the encapsulating dielectric layer on the molecules and the absence of proton source [Singh et al., *Chem. Commun.* **50**, 11204 (2014)].

2. For the charging effect, spectral shifts can be observed by the hot-carrier injection (from the metal to molecules) because it can cause a change in molecular bond lengths [Wang et al., *J. Phys. Chem. C* **124**, 2238 (2020)]. However, in our experiment, we could exclude this hot-carrier injection effect because the tip-molecule distance was maintained large enough (~3 nm) and the dielectric capping layer also suppresses the hot-carrier injection.

3. For the raised concern of the redox reactions, Van Duyne group demonstrated that oxidation-reduction reaction of molecules could be induced by electrochemical TERS [Kurouski et al., *Nano Lett.* **15**, 7956 (2015)]. However, since we did not apply external bias to the BCB molecules, we believe the redox effect was negligible in our observed spectral shifts.

4. Lastly for the possibility of the Stark effect, it was experimentally observed for single molecules sandwiched with the metal nano-gap when the applied DC electric fields at the gap was large enough (~10 V/nm) [Giesecking et al., *J. Phys. Chem. Lett.* **9**, 3074 (2018)]. On the other hand, since we did not apply external bias to the BCB molecules, we believe the Stark effect was negligible in our observed spectral shifts.

[Added in the main text] We exclude other possible effects of the observed spatial shifts for the following reasons. First, spectral shifts can occur depending upon the chemically distinct states, i.e., protonation and deprotonation, of molecules due to the local pH differences [Huang et al., *Sci. Adv.* **7**, eabg1790 (2021)]. However, in our work, the spin-coated BCB molecules are physisorbed on the Au surface (not chemically bound) and the proton transfer process rarely occurs because of the encapsulating dielectric layer on the molecules [Singh et al., *Chem. Commun.* **50**, 11204 (2014)]. Second, spectral shifts can be observed by the hot-carrier injection from the plasmonic metal to molecules because it can cause a change in molecular bond lengths [Wang et al., *J. Phys. Chem. C* **124**, 2238 (2020)]. However, in our experiment, we exclude this hot-carrier injection effect because the tip-molecule distance is maintained at ~3 nm and the dielectric capping layer also suppresses the hot-carrier injection. Third, a previous study demonstrated that oxidation-reduction reaction of molecules could be induced by electrochemical TERS [Kurouski et al., *Nano Lett.* **15**, 7956 (2015)]. However, since we do not apply external bias to the BCB molecules, we believe the redox effect is negligible in our observed spectral shifts. Lastly, the Stark effect was experimentally observed for single molecules sandwiched with the metal nano-gap when the applied DC electric fields at the gap was large enough (~10 V/nm) [Giesecking et al., *J. Phys. Chem. Lett.* **9**, 3074 (2018)]. Since we do not apply external bias to the BCB molecules, we can ignore the Stark effect.

Also, have the authors investigated and eliminated whether any interaction between the BCB and the Al₂O₃ can cause these particular shifts? While I understand that not all alternative explanations can be fully modelled with DFT and compared, the

authors should make a decent effort in considering alternative explanations and include a robust discussion on the subject, supported by either data, modelling or reasoning/justification for elimination.

Spectral shift can be caused by an interaction between the BCB and Al_2O_3 , as the reviewer noted. Therefore, we performed additional DFT calculations for BCB on top of rhombohedral Al_2O_3 , and compared the results with BCB on Au. Our calculation shows that the variance of vibrational energy is much larger for Au than Al_2O_3 . The main reason could be the stronger electronic coupling for metallic Au than insulating Al_2O_3 , as the density of states are presented below. Indeed, we are studying the effect of Al_2O_3 on the BCB molecules as well as the Au tip and the Au substrate more quantitatively, as a follow-up work. Since the simulations are ongoing with various Al_2O_3 conditions, we have a plan to present the well-organized results in the separate paper in the near future. We thank the reviewer for this constructive comment.

Can the authors provide the entire TERS spectrum retrieved for points 5-9? Maybe they can be included in the SI.

We thank the reviewer for pointing this out. We have added the entire TERS spectra for points S4-S9 below in the Supplementary Information.

[Added in the Supplementary information]

Fig. S13. TERS spectra measured at the points S4-S9 (Fig. 4a) in the full spectral range (500 cm^{-1} - 1700 cm^{-1}).

In the introduction, the authors oversell the TERS technique by stating that: "SERS based approaches do not allow single-molecule measurements in heterogeneous chemical environments..." It is not clear what is meant by the heterogeneous environment in this manuscript as only one analyte molecule is probed but in a matrix of Al_2O_3 , but in such an environment measuring single molecules using SERS is certainly possible (see e.g. 10.1038/s41467-021-26898-1).

We agree with the reviewer's opinion. Although single-molecule SERS is generally difficult for heterogeneously distributed molecules on the surface, it is certainly possible. To avoid confusion, we have revised the main text and newly cited several pioneering SERS works as follows:

[Original text] However, SERS-based approaches do not allow single-molecule measurements in heterogeneous chemical environments due to their diffraction-limited spatial resolution. By contrast, TERS gives access to extremely weak vibrational responses of single-molecules and even individual chemical bonds in a single-molecule using a strongly localized optical field at the plasmonic nano-tip, controlled by scanning probe microscopy approaches.

[Revised text] In general, SERS provides a larger enhancement factor in single-molecule detection compared to TERS [Verma et al., *Chem. Rev.* **117**, 6447 (2017)] which gives access to extremely weak vibrational responses of single molecules [Pszona et al., *J. Chem. Phys.* **156**, 014201 (2022)]. On the other hand, TERS allows us to probe even individual chemical bonds in a single molecule using a strongly localized optical field at the plasmonic nano-tip, controlled by scanning probe microscopy approaches.

What step size was used when collecting the TERS images, and has any smoothing, averaging or interpolation been used in the images Fig. 3 a,b and 4a? If so then this should be reported, and raw images shown in the SI.

The step size of the TERS images was 20 nm. For the obtained TERS images, we corrected the background differences of each line along the scanning axis and applied bicubic interpolation. Based on the reviewer's comment, we have added the raw images of Fig. 3a-b in the SI.

[Added in the Supplementary Information]

Fig. S10. Original TERS images (raw data) for $\sim 580\text{ cm}^{-1}$ (a) and $\sim 1160\text{ cm}^{-1}$ peak intensities (b). These images were postprocessed in Fig. 3a-b of the main text as follows: the line defects along the scanning axis were corrected and an interpolation method was used.

Towards the end the authors use the term: "...likewise illustrations in Fig..." This is incorrect and should be revised. The term chosen to replace should be chosen carefully and justified with the aforementioned discussion, as it states the strength/confidence with which the authors claim the geometry in Fig. 5c is the geometry observed in the experiments (based on the match to the Raman spectrum).

We thank the reviewer for pointing this out. To state firmly our interpretation, we have revised the main text as below and we have added a separate paragraph for the reasons why we could exclude other possibilities for the observed spectral shifts as we showed above.

[Original text] From these simulation results, we can deduce that the experimentally observed possible single molecule in S6, S9 (in Fig. 3 and 4) have a chemical environment and molecular orientation likewise illustrations in Fig. 5c.

[Revised text] From these simulation results, we can deduce that the experimentally observed molecules in the spots S6 and S9 (in Fig. 3 and 4) likely have similar molecular orientations to the illustrations in Fig. 5c. The other chemical or environmental conditions give much less effect to the spectral shift compared to the molecular orientation and coupling (mainly C - Au atoms).

Reviewer #2 (Remarks to the Author):

This manuscript reports on tip enhanced hyperspectral Raman imaging of brilliant cresyl blue (BCB) at the single molecule level at room temperature. The carefully

prepared samples consisted of thin, smooth Au-films deposited on low optical background covers slides. A low concentration of BCB molecules deposited on the gold film by spin coating. The sample was covered with a protective capping layer of 0.5 nm Al₂O₃ to suppress spatial and spectral diffusion of the molecules. The optical measurements were performed with an inverted confocal microscope by illuminating the BCB-molecules with a radially polarized beam through the gold film to form a coupled plasmon oscillation between the apex of close sharp gold tip and the gold film right below. By blocking the incident radiation from reaching the detector, images were recorded by raster scanning the sample and detecting the back scattered light from the sample point by point. Due to the low concentration of the BCB molecules on the gold film dim and bright spots with diameters of about 20 nm could be observed in the Stokes shifted part of the spectrum. From these spots Raman spectral of the BCB where recorded. The Raman spectra of nine isolated spots where analyzed. The line widths of the most intense Raman peaks located around 580 cm⁻¹ varied from 7.5 cm⁻¹ to 5.4 cm⁻¹ and their center frequencies differed by as much as 7.5 cm⁻¹. This inhomogeneous behavior indicates that the individual spots may originate from single isolated molecules. Abrupt spectral fluctuation typically observed in single molecule spectroscopy where not observed for the samples with an applied protective layer, however they were observed from unprotected samples and were associated with a considerable spectral line broadening.

We thank the reviewer for summarizing our approach and key outcomes.

Single-molecule TERS is challenging at ambient conditions due to the rapid structural dynamics of the surface bound molecules exposed to the atmosphere. In recent years, TERS-measurements performed in UHV and at cryogenic temperatures in an STM-configuration have excited considerable sensation since they could be performed with very high spatial resolution revealing single atoms in a molecule and single chemical bonds. However, room temperature is required for a large number spectroscopic studies, e.g. to investigate molecular dynamics and functions relevant e.g. for life sciences, biology or energy conversion to name a few. This work is a significant contribution to single-molecule SERS or TERS and has several interesting aspects. 1) It introduces a clever a way to reduce fast dynamics and considerably reduces spectral diffusion or spectral broadening leading to highly resolved Raman spectra at ambient temperatures making it interesting for a wide audience. 2) Raman scattering is performed by exciting through a thin Au-film a coupled plasmon oscillation between the tip and the sample where the vibrational modes of the molecule can perfectly modulate the plasmon oscillation. Hence a strong polarization sensitive Raman spectrum of a single molecule can be detected while at the same time the background of out of focus features are suppressed. This is an interesting and highly welcome alternative to the often used side-illumination scheme. Furthermore, it has to be mentioned that the authors have optimized the

thickness of their Au-films to obtain a strong back-scattering signal from the coupled gap-plasmon mode. 3) Extensive quantum chemical calculations have been performed to explain the heterogeneity in the Raman spectra in terms of the orientation and localization of the molecules with respect to different positions on the Au-film.

This work provides an easy way to investigate the single-molecule properties in interacting molecules at room temperature and hence expands the scope of single-molecule vibrational spectroscopy studies. The experimental approach, the quality of the data and their analysis are valid and the contents is clearly and scholarly presented. I suggest publication after some minor but mandatory revisions that should help to further improve this manuscript.

We sincerely thank the reviewer for acknowledging the novelty and significance of our work. In addition, we thank gratefully the reviewer for describing the advantages of our approach in details. With regard to the raised minor concerns below, we have addressed them clearly and provided supporting results in the point-by-point response below. We have also made corresponding revisions to our manuscript as indicated in red at the advice of the reviewers.

Questions and suggestions to be consider in the review:

- What effect has the Al₂O₃ film on the plasmonic gap field distribution between the tip apex and the gold film?

To clarify the effect of Al₂O₃ film on the plasmonic gap field distribution, we performed FDTD simulations with and without the Al₂O₃ film between the Au nano-gap as shown in the figure below. From these simulations, we confirm that the field distribution at the plasmonic nano-gap is not significantly affected by the existence of Al₂O₃ film.

[Added in the Supplementary Information]

Fig. S9. FDTD-simulated optical field intensity ($|E_z|^2$) distribution at the nano-gap without (a) and with (b) Al₂O₃ film on Au substrate.

To understand the effect of Al₂O₃ film on the optical field distribution at the plasmonic

nano-gap, we performed FDTD simulations with and without the Al₂O₃ film between the Au nano-gap as shown in the figure below. From these simulations, we confirm that the field distribution at the plasmonic nano-gap is not significantly affected by the existence of Al₂O₃ film.

- Does the capping layer also protect the BCB-molecules against photo-bleaching?

Photobleaching of BCB molecules is significantly reduced under vacuum environment compared to the ambient condition because oxygens in air cause the photodecomposition process [Steidtner et al., *Phys. Rev. Lett.* **100**, 236101 (2008)]. Therefore, we do believe the Al₂O₃ capping layer is also helpful to reduce the photobleaching effect. We thank the reviewer for pointing this out and we have added this discussion in the main text.

[Added in the main text] It should be also noted that a previous study demonstrated that photobleaching of BCB molecules is significantly reduced under vacuum environment compared to the ambient condition because oxygen in air causes the photodecomposition process [Steidtner et al., *Phys. Rev. Lett.* **100**, 236101 (2008)]. Hence, the Al₂O₃ capping layer is beneficial to reduce the photobleaching effect of molecules in our experiment.

- Can the authors comment on the effect of the positive charge of the BCB-molecules on the Au-layer and the gap field?

As the reviewer mentioned, BCB molecule is an organic cation with a +1 charge [Kostjukov, *Mol. Phys.* e1996647 (2021)]. The BCB cations will keep their charge as +1 when they are in contact with an Au substrate since the Fermi level of Au is in between the LUMO and HOMO of BCB cation. However, applying bias voltage is necessary to experimentally verify the charge effect of molecules [Braun et al., *J. Am. Chem. Soc.* **141**, 1816 (2021)]. Hence, unfortunately, it is unclear in our experiment that how the charge of BCB molecule effects on the Au layer and the gap field.

- Often TERS spectra of Au-samples probed with an Au tip show a significant background due to interband transitions and inelastic plasmon decay. Why don't we see this background in Fig 1b) ?

Indeed, Au gap mode TERS configuration shows plasmon background because of the Au interband transitions and the plasmon decay [Hennemann et al., *Spectroscopy* **24**, 119 (2010)]. To provide clear information to readers, we have added a spectrum for the tip-enhanced plasmon response on an Au substrate (without BCB molecules) and a full spectrum of Fig. 1b to SI with spectral fittings of

the background response.

[Added in the Supplementary Information]

Fig. S2. Tip-enhanced plasmon response of the Au substrate without BCB molecules (a) and TERS spectrum of BCB molecules on the Au substrate (b). The red solid lines are experimental data and the blue dashed lines are background fit with Voigt function.

- The radial laser mode gives a dominant longitudinal field component in the center which is wanted to excite a strong gap plasmon-polariton field and allows for polarization sensitive measurements. My question points to the azimuthally polarized ring which surrounds the optical axis. For a 1.3 NA objective lens this field intensity component is still significant and increases the focal diameter and probably contributes to the PL background from Au. On the other hand it can efficiently be scattered from horizontal molecule revealing in-plane vibrational modes. Can the authors comment on this issue and say why they have not used a higher NA? Hence, it would be very worthwhile to also add the calculated radial field components either along with Fig. 2b and Fig. 2c or add them to the SI.

We do generally agree with the reviewer's opinion that we can use the azimuthally polarized beam to selectively probe in-plane vibrational mode of molecules. However, while this clever idea is very effective in far-field measurements [Chizhik et al., *J. Phys. Chem. Lett.* **2**, 2152 (2011)], we have to carefully assess the feasibility in near-field measurements because the metallic tip and substrate can significantly modify the polarization state at the nano-gap [Park et al., *Nano Lett.* **18**, 2912 (2018)]. We thank the reviewer for providing excellent idea for the follow-up work and we would like to try regarding simulations and experiments systematically in the near future.

For the reviewer's suggestion of displaying the calculated radial field component, we have added the optical field distributions at the nano-gap for the vertical and horizontal field components separately in the SI when the excitation light is radially polarized beam.

[Added in the Supplementary Information]

Fig. S5. FDTD-calculated optical field intensity distributions at the plasmonic nano-gap for vertical (a) and horizontal (b) field components with the total field intensity (c).

Since the polarization state at the plasmonic nano-gap can affect to the probing vibrational modes of a molecule [Chizhik et al., *J. Phys. Chem. Lett.* **2**, 2152 (2011)], we provide the optical field distributions at the nano-gap for the vertical and horizontal field components separately when the excitation light is radially polarized beam [Khoptyar et al., *Opt. Express* **16**, 9907 (2008)], as shown in Fig. S5. This result clearly shows that the vertical component is dominant at the plasmonic nano-gap and we can selectively probe the out-of-plane vibrational modes in our experiment.

- A gap mode created between a smooth gold sample and a sharp gold tip excited by radial polarization has been used for tip enhanced Raman and luminescence spectroscopy before. An article closely related to this work that should be cited is “Imaging nanometre-sized hot spots on smooth Au films with high-resolution tip-enhanced luminescence and Raman near-field optical microscopy” by M Sackrow et al, *ChemPhysChem* **9** (2), 316-320, 2008.

We thank the reviewer for the reference suggestion. We have newly cited the Sackrow’s paper in the revised manuscript.

- I do not agree with the following statement: “However, SERS-based approaches do not allow single-molecule measurements in heterogeneous chemical environments due to their diffraction-limited spatial resolution.” Single-molecule measurements based on SERS in heterogeneous chemical environments have been shown and investigated in detail by several authors, exhibiting inhomogeneous dynamic spectral behavior due to the heterogeneous nature e.g. of the Ag-aggregate, although diffraction limited. The authors should formulate their statement more clearly and should also cite some of the pioneering single-molecule SERS references, which show such spectral fluctuations e.g. S. Nie and S. R. Emory, *Science* **275**, 1102–1106 (1997). K. Kneipp, et al., *Phys. Rev. Lett.* **1997**, **78**, 1667– 167, T Vosgröne, AJ Meixner, *ChemPhysChem* **6** (1), 154-163, 2005. I recommend tha the authors check out a very recent SERS- article by M Pszona et al. *J Chem Phys* **156**, 014201, 2022.

We agree with the reviewer’s opinion. Although single-molecule SERS is generally difficult for heterogeneously distributed molecules on the surface, it is certainly

possible. To avoid confusion, we have revised the main text and newly cited several pioneering SERS works as follows:

[Original text] However, SERS-based approaches do not allow single-molecule measurements in heterogeneous chemical environments due to their diffraction-limited spatial resolution. By contrast, TERS gives access to extremely weak vibrational responses of single-molecules and even individual chemical bonds in a single-molecule using a strongly localized optical field at the plasmonic nano-tip, controlled by scanning probe microscopy approaches.

[Revised text] In general, SERS provides a larger enhancement factor in single-molecule detection compared to TERS [Verma et al., *Chem. Rev.* **117**, 6447 (2017)] which gives access to extremely weak vibrational responses of single molecules [Pszona et al., *J. Chem. Phys.* **156**, 014201 (2022)]. On the other hand, TERS allows us to probe even individual chemical bonds in a single molecule using a strongly localized optical field at the plasmonic nano-tip, controlled by scanning probe microscopy approaches.

In addition, based on the reviewer's suggestion, we have newly cited pioneering SERS works [works of Nie, Emory, and Meixner's groups] in the first sentence of the introduction.

Reviewer #3 (Remarks to the Author):

This work from the Park group is nicely presented with some beautifully performed experiments supported by numerical simulations. The authors have claimed to use the "freeze-frame" approach by trapping the sample molecules under a thin layer of aluminum oxide and have shown TERS images of BCB molecules. I have a few concerns, as mentioned below, and based on them, I suggest a minor revision of this paper before it can be accepted for publication in Nature Communications.

We sincerely thank the reviewer for acknowledging the novelty our work with high praise. With regard to the raised minor concerns below, we have addressed them clearly and provided supporting results in the point-by-point response below. We have also made corresponding revisions to our manuscript as indicated in red at the advice of the reviewers.

The authors claim to have imaged single molecules in this work. Although they slightly toned down the claim by adding the word "possibly" and say that they have demonstrated hyperspectral TERS imaging of "possibly single molecule" at room temperature. Although I highly rate this work, I am not satisfied with the claim of a single molecule, even with the toned-down claim. The authors need to provide more convincing evidence for possible single molecule observation.

We thank the reviewer for pointing this out. We agree with the reviewer's comment, but it was hard to provide the direct single-molecule evidence experimentally in our approach. In previous room temperature single-molecule TERS/SERS experiments, a probability histogram from the time-dependent spectral fluctuation was used as a single-molecule evidence. However, since we obtained stable non-fluctuating time-dependent TERS spectra due to dielectric capping layer, we cannot demonstrate that. As an alternative way, we focused on preparing the low molecular-density sample and encapsulated them with a thin dielectric layer. To provide the evidence of single-molecule observation, we modeled single BCB molecules on the Au surface with various conditions (molecular orientation and substrate-molecule coupling) and simulated the normal modes with DFT calculations. From these quantitative spectral analyses (Fig. 5), we could provide the reasonable evidence of single molecules and probe the conformational heterogeneity of them at room temperature. Nevertheless, since we have not provided direct single-molecule evidences like previous cryogenic STM-TERS experiments [Zhang et al., *Nature* **498**, 82 (2013), Lee et al., *Nature* **568**, 78 (2019)], we have described our experimental results as observations of single or a few molecules or possibly from single molecules in the manuscript. But, fortunately, from the reviewer's constructive comments and our further careful consideration for our results below, we believe that our revised manuscript provides convincing spectral features of single or a few molecules at room temperature.

First, TERS has two features, high spatial resolution and high field enhancement. Both or either can be used for single molecule detection. That means, either the spatial resolution should be extremely high to distinguish every molecule individually, or the sample should be prepared in such a diluted way that molecules are isolated and only one molecule is present in the observation volume. In this work, the authors have not said anything about the spatial resolution, but one can see from Figure 3a that it is about 20 nm, which is a typical value in TERS imaging. With such spatial resolution, the only possibility is to prepare an extremely diluted sample to make sure molecules are isolated, which seems to be the case here in this paper. Once such a sample is prepared, there is nothing much that TERS does, except locating the molecule and enhancing the signal to a level that a single molecule can be observed. Such enhancement factor should be at least a million times, if not more.

The reviewer understands our approach correctly. Due to the drifts of sample and tip originated from thermal expansions in ambient condition, we have not focused to reduce the spatial resolution in TERS imaging. Instead, we prepared extremely diluted and well isolated BCB molecules and detected them with large TERS enhancement factor as high as $\sim 10^5$ -fold.

Is the aim of the paper related to high enhancement in TERS? However, the authors have not discussed anything about the enhancement factor. As I mentioned, once a

good sample is prepared, TERS does not have much role to play other than huge enhancement. So my first suggestion is that the authors should estimate the enhancement factor and discuss if it is sufficient for the observation of a single molecule or not. Although, I do believe that the enhancement is sufficiently high, otherwise authors will not observe such a sample by TERS. Or, if the goal of this work is something other than what I have mentioned, then the authors should clearly mention that.

We agree that we need to present our TERS enhancement factor and discuss the feasibility of single-molecule detection in the manuscript. From TERS measurement of molecular ensembles (Fig. 1b), we could estimate the TERS enhancement factor of $\sim 10^5$ (Note that the calculation of enhancement factor for single-molecule TERS spectra was not possible because the far-field Raman scattering gave no signal). We have added the discussion for the TERS enhancement factor in the revised text with the details of calculation method in the SI.

[Added in the main text] Through the optimization process, the estimated TERS enhancement factor in our experiment is $\sim 2.0 \times 10^5$ (see Supplementary Information for details), which is sufficient for single-molecule level Raman scattering detection as discussed in the previous study [Park et al., *Nano Letters* **10**, 4040 (2010)].

[Added in the Supplementary Information]

Calculation of TERS enhancement factor

We use the following equation for empirical estimation of TERS enhancement for our BCB experiment [Stadler et al., *Nano Lett.* **10**, 4514 (2010)]:

$$EF = \left(\frac{I_{tip-in} - I_{tip-out}}{I_{tip-out}} \right) \times \frac{A_{FF}}{A_{NF}}$$

where I_{tip-in} is the TERS intensity and $I_{tip-out}$ is the Raman intensity with the tip fully retracted (far-field) from the sample surface. A_{FF} refers to the focused beam spot by an oil-immersion objective lens with $NA = 1.30$, and A_{NF} refers to near-field excitation region of the Au tip-Au substrate. I_{tip-in} and $I_{tip-out}$ values are derived from the curve fitting by Voigt function, and the values of A_{FF} and A_{NF} derived using a general equation for an area of a circle πr^2 . The far-field laser spot radius is $r_{FF} \approx \left(\frac{\lambda}{2NA} \right) \times 1.5 = \sim 365 \text{ nm}$ (the factor 1.5 is the empirical factor) [Neacsu et al., *NanoBiotechnol.* **3**, 172 (2007)] and the near-field radius is $r_{NF} = 10 \text{ nm}$, which is a half of the spatial resolution of the Au tip. Using the equation above and our experimental results in Fig. 1b, estimated TERS enhancement factor is as high as $\sim 2.0 \times 10^5$.

My second point is about how the authors distinguish between single and multiple

molecules in Fig. 4, which is later supported by simulation in Fig. 5. The authors claim that spots 1-4 with small shifts in peak are multiple molecules while spots 5-9 with large shifts in peak are single molecules. This is not very clear to me. If there are multiple molecules, and if they have different conformations (for example, one molecule has conformation as in Fig. 5b and another as in Fig. 5d), one would effectively see a large peak shift depending upon the individual conformations. Also, there will be effectively large peak widths for the same reason. This would then be the case of Fig. 4d rather than Fig. 4c. Moreover, Figs. 3a, 3c and 4a clearly show that spots 5, 7 and 8 do not represent a single molecule because these points have large spread (up to 100 nm in size in the x-direction) in the images. Considering the spatial resolution to be around 20 nm, this is possible only if multiple molecules are aligned along the x-direction. Or is there something that I am missing here? The authors should clarify how they distinguish single and multiple molecules, because their current explanation is not very clear.

We agree with the reviewer that distinguishing single and multiple molecules was unclear in the previous manuscript. As the reviewer mentioned, multiple or a few molecules can have different conformations, and thus a large peak difference can also be observable. To show the experimental evidence, we have added an additional TERS image for a few molecules with a TERS spectrum in the SI (please see below), which supports the reviewer's expectation (large peak difference of a few molecules due to the conformational heterogeneity).

[Added in the Supplementary Information]

Fig. S14. (a) TERS peak intensity image of BCB molecules for the vibrational mode at $\sim 580 \text{ cm}^{-1}$. (b) TERS spectrum obtained from the red arrow indicated in (a).

Fig. S14a and b show the TERS peak intensity image of BCB molecules for the vibrational mode at $\sim 580 \text{ cm}^{-1}$ with TERS spectrum obtained from the red arrow indicated in Fig. S14a. The observed TERS spot is expected to be a few molecules from the consideration of the spatial distribution, peak intensity, and linewidth of the TERS spectrum. From the simple curve fits of the TERS spectrum in Fig. S14b, we can clearly confirm the conformational heterogeneity of a few molecules.

In addition, after careful considerations, we have re-separated the observed spots 1-9 into single/a few/multiple molecules as follows: S1-S3 are multiple (strong response with broad linewidth) and S4-S9 are single or a few (relatively weak response with relatively narrow linewidth). Specifically, S4 is a few (broad linewidth), S5 is a few (spatially spread out), S6 is single (narrow linewidth and spatially isolated), S7 is a few (spatially spread out), S8 is a few (spatially spread out), and S9 is single (narrow linewidth and spatially isolated). The main change in the revised manuscript is the spot S4. Since we verified that a few molecules can show a broad TERS linewidth (Fig. S14), from considerations of the peak intensity and the spatial distribution, we could conclude that the S4 should be classified in the a few molecules group. Based on this new classification, we have revised the main text including Fig. 4.

[Revised Fig. 4]

Fig. 4 (a) TERS peak intensity image of the vibrational mode at $\sim 580 \text{ cm}^{-1}$ of BCB molecules. (b) TERS spectra measured at spots 1, 2 and 3 indicated with red circles in (a). TERS spectra measured at spots 1-3 (c) and spots 4-9 (d) fitted with Voigt line shape function in the range from 565 cm^{-1} to 595 cm^{-1} . From the observed TERS peak shift in Fig. 4d, the TERS spectra at spots 1-3 (red circles in Fig. 4a) are probably measured from molecular ensembles and spots 4-9 (dark blue circles in Fig. 4a) are possibly measured from single or a few molecules. In Fig. 4c and d, the dots and lines are experimental data and fitted curves, and the TERS spectra at each spot are also shown as background 2D contour images.

[Revised main text]

Further, to distinguish the single, a few, or multiple molecules, we also need to consider the spatial distribution in TERS images in addition to spectroscopic information. Hence, we analyze the spectral properties and spatial distribution of the observed spots in the TERS image to obtain the evidence of single-molecule detection. We classify the observed TERS spots in Fig. 3 into two groups, as shown

in Fig. 4a. We surmise that the TERS response in the first group (red circled spots 1-3) was measured from multiple BCB molecules because the TERS signal of both the $\sim 580\text{ cm}^{-1}$ and the $\sim 1160\text{ cm}^{-1}$ modes are pronounced, significant TERS peak shift is not observed, and the peaks have generally broad linewidths, as shown in Fig. 4b and c (see also Fig. 3a and b). By contrast, we observe much weaker TERS responses in the second group (blue circled spots 4-9 in Fig. 4a) with a significant peak variation corresponding to $\sim 580\text{ cm}^{-1}$, as large as $\sim 7.5\text{ cm}^{-1}$, as shown in Fig. 4d (see Fig. S13 for the full spectral range). We then consider the linewidth of each spectrum and spatial distribution in the TERS image to distinguish the single and a few molecules. First of all, we classify the spot 4 as a few molecules. Although its TERS intensity is weak with the small spatial distribution, the linewidth is quite broad (7.5 cm^{-1}) comparable to the first group (see Table S2 for details). It should be noted that a few molecules can show broad linewidth even with double peaks when the molecules have different conformations (see Fig. S14 for details). The remaining five spots (S5-9) show narrow linewidths of $\leq 6.0\text{ cm}^{-1}$, which is closer to the spectral resolution in our experiment (see also Fig. S12). Hence, we finally classify these spots into single or a few molecules group based on the spatial distribution in TERS image. As can be seen in Fig. 4a, the TERS response of spots S5, S7, and S8 is spatially spread out, which means the signals were obtained from a few molecules. Therefore, from the spatio-spectral analyses, we believe the TERS responses from S6 and S9 are possibly originated from single isolated BCB molecules.

In addition to my concerns mentioned above, I have a few minor points-

1. Overall, the paper is written in a clear way, but there are some issues with the language here are there. The authors should pay more attention to the language.

We appreciate the reviewer for the constructive comment. We have revised the whole manuscript for correct expression in English by the help of a native speaker.

2. The idea of trapping the sample molecules under a thin layer of aluminum oxide to restrain small fluctuation in position or orientation of sample molecules at room temperature is very nice. However, the authors have not mentioned if this capping also affects the molecular vibration or not? If it does, has it been taken into consideration? Although, even if the molecular vibrations are altered, it will not affect the results of this work.

We thank the reviewer for this constructive comment. Spectral shift can be caused by an interaction between the BCB and Al_2O_3 . Therefore, we performed additional DFT calculations for BCB on top of rhombohedral Al_2O_3 , and compared the results with BCB on Au. Our calculation shows that the variance of vibrational energy is much larger for Au than Al_2O_3 . The main reason could be the stronger electronic

coupling for metallic Au than insulating Al_2O_3 , as the density of states are presented below. Indeed, we are studying the effect of Al_2O_3 on the BCB molecules as well as the Au tip and the Au substrate more quantitatively, as a follow-up work. Since the simulations are ongoing with various Al_2O_3 conditions, we have a plan to present the well-organized results in the separate paper in the near future.

3. The authors mention that the aluminum oxide layer on the sample molecule prevents “unnecessary contamination” of the tip. Although I understand what the authors mean and it is also explained through Fig. S1, it is not the usual way to refer to tip contamination in TERS. Since the experiment is performed in ambient conditions, there are many possibilities for the tip to get contaminated with foreign molecules. Carbon and sulfur are the most common contaminations during TERS experiments, which come from the atmosphere. Instead of simply using the word “contamination”, the authors should instead clearly mention that they could prevent the sample molecules from getting picked up by the tip during the imaging process. Also, it is incorrect to say that the thin layer “protects plasmonic tip”! It rather protects the sample molecules.

We thank the reviewer for pointing out the misleading sentence. Since we totally agree with the reviewer’s comment, we have revised the sentence as follows in the revised manuscript.

[Original text] The capping layer provides a freeze-frame for individual molecules, and it protects plasmonic tip and metal surface from unnecessary contaminations and chemical interactions.

[Revised text] The capping layer not only provides a freeze-frame for individual molecules, but also protects them from unwanted chemical contamination under

ambient conditions, especially by sulfur [Worley et al., *J. Vac. Sci. Technol.* **13**, 2281 (1995)] and carbon [Comini et al., *J. Vac. Sci. Technol.* **39**, 043203 (2021)] molecules. Moreover, it prevents possible contamination of the Au tip, e.g., adsorption of the probing molecules on to the tip surface that can cause artifact signals, as shown in Fig. 1a (see Fig. S1 for more details).

4. When a metallic tip is used in combination with a metallic substrate, one needs to consider not only the transmission of light through the substrate, but also the possible hybridization of plasmons, which affects the enhancement at a given wavelength. The authors should refer to a previous paper (Nanoscale, vol. 4, p. 5931, 2012), where a similar result was shown for a similar wavelength. The reference paper demonstrated that for 642 nm, the best enhancement in TERS was obtained when the substrate was 12 nm thick.

We thank the reviewer for recommending this highly relevant literature to our work. We have cited this paper in the revised manuscript.

[Added in the main text] Note that a previous study demonstrated that the plasmon resonance from a ~12 nm thick Au film gives rise to the largest TERS enhancement for the optical responses at ~640 nm [Uetsuki et al., *Nanoscale* **4**, 5931 (2012)].

4. The reference of Fig. S3 comes after that of Fig. S5 in the main text. Please take care of the sequence.

We apologize for our mistake. We have corrected the sequence of the SI figures.

5. The intensity ratio in Fig. 3c is not clear. Should the ratio depend on the molecule number or the conformation of the molecule, or should it be uniform? What is the reason that some spots are much stronger than others? For example, spots 5, 7 and 8, which are supposed to be single molecules, have much more intensity than spots 1, 3, 4, which are supposed to be multiple molecules.

Fig. 3c shows the peak intensity ratio of ~580 cm⁻¹ vibrational mode (Fig. 3a) and ~1160 cm⁻¹ vibrational mode (Fig. 3b). We believe the displayed variation of the peak intensity in Fig. 3c is mainly originated from the conformational heterogeneity of molecules in space. Because the dominant vertical field component under the TERS tip selectively probes the out-of-plane vibrational modes with respect to the surface. Under this circumstance, we expect that the peak intensity ratio can be low for the multiple molecules in comparison with the peak intensity ratio of single molecules. We have added a note in the revised manuscript.

[Added in the main text] The conformational heterogeneity of the probed molecules can be best exemplified by the peak intensity ratio of the vibrational modes at ~580 cm⁻¹ (Fig. 3a) and at ~1160 cm⁻¹ (Fig. 3b), as shown in Fig. 3c. Since TERS tip

selectively probes the out-of-plane modes with respect to the surface, the peak intensity ratio should be low for multiple molecules in comparison with the single molecules.

Reviewer comments, second round review

Reviewer #1 (Remarks to the Author):

The authors have addressed all my comments and concerns.
I suggest publish as is.

Reviewer #2 (Remarks to the Author):

The authors have carefully answered the reviewers questions and comments and made appropriate changes to their manuscript. The outcome is a very nice scholarly research article which I highly recommend for publication in the present form.

Reviewer #3 (Remarks to the Author):

The authors have satisfactorily and clearly answered all my concerns and have accordingly updated the revised manuscript. Now this manuscript presents a wonderful piece of research and I recommend its publication in the present form.